Recommended survey designs for occupancy modelling using motion-activated cameras: insights from empirical wildlife data

Shannon Graeme 1 2 graeme.shannon@colostate.edu
Lewis Jesse S. 1 3
Gerber Brian D. 4
1 Department of Fish, Wildlife, and Conservation Biology, Colorado State University , Fort Collins, CO , USA
2 National Park Service, Natural Sounds and Night Skies Division , Fort Collins, CO , USA
3 Graduate Degree Program in Ecology, Colorado State University , Fort Collins, CO , USA
4 Colorado Cooperative Fish and Wildlife Research Unit, Department of Fish, Wildlife and Conservation Biology, Colorado State University , Fort Collins, CO , USA
Gandini Patricia
Electronic publication date: 2014 Aug 28
Publication date: 2014
Volume: 2
Electronic Location ID: e532
Received 2014 Jun 16; Accepted 2014 Aug 1
Copyright: © 2014 Shannon et al.
Copyright year: 2014
Copyright holder: Shannon et al.
License: This is an open access article distributed under the terms of the Creative Commons Attribution License, which permits unrestricted use, distribution, reproduction and adaptation in any medium and for any purpose provided that it is properly attributed. For attribution, the original author(s), title, publication source (PeerJ) and either DOI or URL of the article must be cited.
License URL: https://creativecommons.org/licenses/by/4.0/

Keywords: Animal, Conservation, Detection probability, Landscape, Mammal, Monitoring, Sampling, Simulation, Species distribution, Survey effort

Funding: NSF NSF EF-0723676 Colorado State University National Park Service Funding for the camera study was provided by a grant from the National Science Foundation (NSF EF-0723676). Colorado State University provided postdoctoral funding for GS in partnership with the National Park Service. The funders had no role in study design, data collection and analysis, decision to publish, or preparation of the manuscript.

==============================
Motion-activated cameras are a versatile tool that wildlife biologists can use for sampling wild animal populations to estimate species occurrence. Occupancy modelling provides a flexible framework for the analysis of these data; explicitly recognizing that given a species occupies an area the probability of detecting it is often less than one. Despite the number of studies using camera data in an occupancy framework, there is only limited guidance from the scientific literature about survey design trade-offs when using motion-activated cameras. A fuller understanding of these trade-offs will allow researchers to maximise available resources and determine whether the objectives of a monitoring program or research study are achievable. We use an empirical dataset collected from 40 cameras deployed across 160 km2 of the Western Slope of Colorado, USA to explore how survey effort (number of cameras deployed and the length of sampling period) affects the accuracy and precision (i.e., error) of the occupancy estimate for ten mammal and three virtual species. We do this using a simulation approach where species occupancy and detection parameters were informed by empirical data from motion-activated cameras. A total of 54 survey designs were considered by varying combinations of sites (10–120 cameras) and occasions (20–120 survey days). Our findings demonstrate that increasing total sampling effort generally decreases error associated with the occupancy estimate, but changing the number of sites or sampling duration can have very different results, depending on whether a species is spatially common or rare (occupancy = ψ) and easy or hard to detect when available (detection probability = p). For rare species with a low probability of detection (i.e., raccoon and spotted skunk) the required survey effort includes maximizing the number of sites and the number of survey days, often to a level that may be logistically unrealistic for many studies. For common species with low detection (i.e., bobcat and coyote) the most efficient sampling approach was to increase the number of occasions (survey days). However, for common species that are moderately detectable (i.e., cottontail rabbit and mule deer), occupancy could reliably be estimated with comparatively low numbers of cameras over a short sampling period. We provide general guidelines for reliably estimating occupancy across a range of terrestrial species (rare to common: ψ = 0.175–0.970, and low to moderate detectability: p = 0.003–0.200) using motion-activated cameras. Wildlife researchers/managers with limited knowledge of the relative abundance and likelihood of detection of a particular species can apply these guidelines regardless of location. We emphasize the importance of prior biological knowledge, defined objectives and detailed planning (e.g., simulating different study-design scenarios) for designing effective monitoring programs and research studies.

Introduction

Estimating the distribution of a species or suite of species across the landscape provides wildlife biologists with crucial information for monitoring and conserving animal populations (Noon et al., 2012). It is also a key criteria for global conservation initiatives such as the International Union for Conservation of Nature red list (http://www.iucnredlist.org/), which has been used to track the change in extinction risk of threatened species over time (Di Marco et al., 2014). Motion-activated cameras are one of the fastest growing techniques for surveying a wide range of terrestrial animals, particularly those that are rare, elusive or cryptic (O’Connell, Nichols & Karanth, 2011; Jamie, 2012). The advancement of affordable and reliable digital camera technology in combination with infrared triggers and time delays has enabled biologists to deploy multiple cameras simultaneously to collect data in an efficient and minimally invasive manner. These data have allowed biologists to investigate a diversity of ecological and conservation driven questions, relating to species abundance (Gerber et al., 2010) and density (O’Brien & Kinnaird, 2011), animal behaviour (Maffei et al., 2011), survival (Gardner et al., 2010), temporal activity (Ridout & Linkie, 2009), and landscape-level occurrence (Thorn et al., 2009). Cameras are typically more efficient than traditional sampling methods (e.g., direct observation, radio telemetry) as continuous data can simultaneously be collected on multiple species (e.g., large bodied carnivores; O’Brien & Kinnaird, 2011). The field deployment can be standardised and readily replicated, enabling researchers to monitor whether there are changes in the occurrence of target indicator species over both time and space (Ahumada et al., 2011; Ahumada, Hurtado & Lizcano, 2013).

Occupancy modelling, which uses detection/non-detection data to estimate species occurrence, offers a very useful analytical framework for analysing data collected from motion-activated cameras (O’Connell & Bailey, 2011). Occupancy models explicitly recognize that given a species occurs in an area, the probability of detecting it on a single survey is often less than one. This potential source of bias is addressed by using repeat sampling across multiple sites, enabling detection probability to be calculated and incorporated in the occupancy estimate (MacKenzie et al., 2002; MacKenzie et al., 2006). Among the key benefits of occupancy studies is that detection/non detection data can generally be collected with greater ease and cost effectiveness for a greater number of species than the more detailed demographic data that are commonly required for estimates of abundance and density (Jones, 2011). As a result, occupancy modelling is increasingly used to evaluate species distribution (Long et al., 2010), habitat use (Betts et al., 2008) and population dynamics (MacKenzie et al., 2010). The results from these studies and monitoring programs have the potential to be used by wildlife managers and conservation practitioners to determine changes in the distribution of key animal populations as well as strengthening future demographic predictions (Jones, 2011; Noon et al., 2012).

There are clear advantages to using motion-activated cameras in occupancy studies; nevertheless, in common with other survey techniques, the efficacy of these studies and monitoring programs relies on appropriate and detailed survey design. These considerations include deciding upon what time period to sample, the sampling length, and the number of cameras to deploy, which is dependent on the target species and the type of inference that is sought (MacKenzie et al., 2006). For community studies, it is important to recognize that an optimal survey for one species may not be so for another; designing a community-level occupancy study will likely incur trade-offs in efficiency and the scope of inference depending on how well the sampling period and duration coincides with a meaningful biological time frame for each species. Research studies and monitoring programs that are initiated without well-defined objectives and rigorous survey design increase the likelihood of returning results that are insufficient to make meaningful inference on the species or system of interest (Yoccoz, Nichols & Boulinier, 2001; Kéry & Schmid, 2004; Mattfeldt, Bailey & Grant, 2009). Moreover, as conservation and research programs are often limited by the availability of funding, it is crucial that surveys are justified in terms of the costs and benefits of acquiring the data (Nichols & Williams, 2006; McDonald-Madden et al., 2010).

Estimates of population parameters often require spatial and temporal replication; in occupancy studies this generates a trade-off in survey effort between the number of sites to sample and the number of replicates to conduct at each site (MacKenzie et al., 2002; MacKenzie et al., 2003; Tyre et al., 2003; MacKenzie et al., 2006). A further consideration is that occupancy is assumed to be static during the designated sampling period (assumption of closure; MacKenzie et al., 2002), and the length of this period may vary depending upon the species and biological timeframe of interest (e.g., the breeding season; Webber, Heath & Fischer, 2013). There are a number of studies that provide theoretical background to study design using an occupancy-modelling framework, highlighting the importance of balancing temporal and spatial replication to most efficiently achieve defined objectives (MacKenzie & Royle, 2005; Bailey et al., 2007; Guillera-Arroita, Ridout & Morgan, 2010; Guillera-Arroita & Lahoz-Monfort, 2012). However, there are few studies that have used empirical data from a suite of species to evaluate the effects of varying the number of sites and occasions on the accuracy and precision of occupancy estimates. Moreover, the majority of research exploring the effective use of motion-activated cameras has focussed on (1) comparing cameras with other sampling approaches (Rovero & Marshall, 2009; Janečka et al., 2011), (2) investigating sampling efficiency as a function of biological parameters (e.g., species, sex, habitat, and season; Larrucea et al., 2007; Kelly & Holub, 2008), and (3) evaluating alternative approaches to species inventories (Tobler et al., 2008; Si, Kays & Ding, 2014). However, a recent analysis conducted on a dataset of avian and mammalian scavengers in sub-arctic environments provided the first detailed discussion of guidelines to determine the optimal survey design for estimating occupancy using empirical data collected from time-triggered cameras (Hamel et al., 2013).

The aim of this study is to provide practitioners who may not have expertise in study design, ecological modeling, or statistics with detailed examples of how survey design influences the accuracy and precision (i.e., error) of occupancy estimates across a range of mammal species using an extensive motion-activated camera dataset. Empirical data allows us to make the tangible connection between values of detection probability, occupancy and the outcomes of the survey design analysis for a given species. In addition, occupancy and detection values for species targeted in motion-activated camera studies (e.g., small-large bodied mammals), as well as sampling length are often in combinations that are not typically evaluated in other study design papers. In fact it is common that per occasion, detection probability is often much lower than values that are usually evaluated (< 0.2 or even <0.1). Furthermore, we envisage that the survey designs that we test in this paper can apply to research that uses motion-activated cameras to study taxa across a diverse range of terrestrial ecosystems.

Our specific research objective was to evaluate how varying the number of sampling sites (10–120 cameras) in combination with the number of occasions (20–120 survey days) influences the error associated with estimating occupancy for 10 mammal species and three ‘virtual’ species. These thirteen species characterize a range of comparatively rare-to-common species with low-to-moderate detection probability, which are typically encountered during camera sampling of terrestrial mammals. Using these results, we provide recommendations and general guidelines that can be used by wildlife practitioners to design and implement studies to evaluate mammal occurrence using motion-activated cameras.

Methods

Study site

The study site was located on the Western Slope (WS) of Colorado, USA on the Uncompahgre Plateau near the towns of Montrose and Ridgway (Fig. 1). The area was characterized by mesas, canyons, and ravines, with elevations ranging from 1800 m to 2600 m and annual precipitation of 430 mm arriving primarily from winter snows and summer thunderstorms (NOAA National Climatic Data). The vegetation communities were dominated by pinyon pine (Pinus edulis), juniper (Juniperus osteosperma), ponderosa pine (Pinus ponderosa), aspen (Populus tremuloides), gambel oak (Quercus gambelii), and big sagebrush (Artemesia tridentata). The WS had extensive areas of undeveloped natural habitat managed by the Bureau of Land Management, US Forest Service, and private landowners. Paved and unimproved roads occurred throughout the WS. The WS has a history of ranching with some private ranches converted into exurban and rural housing developments.

Figure 1 Location of the study site on the Western Slope, Colorado, USA.

The camera survey was completed in 2009 across 40 grid cells covering 160 km2, with one camera per cell.

Study design

We deployed 40 motion-activated cameras across two survey grids totaling 160 km2, with individual camera sites spaced approximately 2 km apart. The sampling design was specifically focused on surveying mountain lions (Puma concolor) and bobcats (Lynx rufus) with cameras placed along game trails, hiking trails, and secondary dirt roads. The placement of cameras along likely travel routes of mammals is common in camera studies and often leads to detecting a diverse assemblage of the mammalian community (O’Connell, Nichols & Karanth, 2011). We checked cameras approximately every two weeks to replace memory cards and batteries if required. The sampling approach was passive in that we did not use attractants (i.e., sight, sound, scent) to lure animals to the camera location. Motion-activated cameras operated from August 21 to December 13, 2009. As the study involved non-invasive sampling using motion-activated cameras there was no requirement for institutional review of the proposed research. Data collection was funded by a grant from the National Science Foundation (NSF EF-0723676).

Data and statistical analyses

We took a two-step approach in our analyses. First, the empirical data collected from motion-activated cameras were used to estimate daily detection probabilities and occupancy estimates for a range of terrestrial mammal species with closure assumed for the entire sampling period (i.e., no changes in occupancy). Second, this information was used in simulations to evaluate optimal survey design approaches for the different species. Photographic data were analysed for ten mammal species (see Fig. 2; the number of photographs are provided in parentheses), raccoons (Procyon lotor: 8), spotted skunks (Spilogale putorius: 25), mountain lions (83), black bears (Ursus americanus: 96), gray foxes (Urocyon cinereoargenteus: 144), coyotes (Canis latrans: 192), elk (Cervus canadensis: 196), bobcats (225), cottontail rabbits (Sylvilagus nuttallii: 1267) and mule deer (Odocoileus hemionus: 1753).

Figure 2 Motion-activated camera images of mammal species included in the study.

(A) Raccoon, (B) spotted skunk, (C) elk, (D) mountain lion, (E) coyote, (F) bobcat, (G) gray fox, (H) black bear, (I) mule deer and (J) cottontail rabbit (low to high detection probability).

For the purpose of our study, a sampling occasion was defined as a 24 h period, which we refer to as a survey day. However, values of detection probability are dependent upon the length of the sampling occasion and researchers will often employ sampling occasions that are measured in weeks rather than days (Ellis, Ivan & Schwartz, 2013). Thus, if the daily detection probability is 0.03, we can recalculate p using the p* formula (p* = 1−(1−p)s) such that at 1 week p = 0.19, 2 weeks = 0.35 and so on (s is the number of sampling occasions). It is also important to note that optimal design for continuous sampling protocols has also recently been explored, which does not rely on discretizing the data (Guillera-Arroita et al., 2011).

Species–specific detection histories were generated for each of the 40 cameras across the four-month sampling period (except black bears, where only the first two months of data were used due to animals hibernating in November and December). For a given species, detection histories provide a record of whether the species was detected (1) or not detected (0) on each survey day for each camera location (40 detection histories for each species). These detection histories were then used to estimate a constant occupancy (ψi) and constant detection probability (pi) for each species i from i = 1, 2, …, 10 using the single-species, single-season occupancy model (MacKenzie et al., 2002). In addition, we created three ‘virtual’ species that were not characterized by our empirical data, but that researchers might encounter, to provide examples where daily detection probability is relatively high (>0.1), while occupancy levels are low to moderate (≤0.6; Fig. 3). We constructed models using the RMark package (Laake & Rexstad, 2014) in the R programming language (R Development Core Team, 2011), which interfaces with Program MARK (White & Burnham, 1999). The resulting 13 species provide a range of daily detection probabilities and occupancy estimates that are typical for mammals surveyed with motion-activated cameras. The species were classified into seven distinct groups ranging from rare and hard to detect species (i.e., raccoon and spotted skunk) to common detectable species (i.e., cottontail rabbit and mule deer; see Fig. 3).

Figure 3 Occupancy estimates and detection probability for 10-mammals and three virtual species that we used to investigate sampling design trade-offs in a simulation exercise.

The species are grouped according to common characteristics: (A), low occurrence and low detection probability; (B), moderate occurrence and low detection probability; (C), high occurrence and low detection probability; (D), moderate occurrence and moderate detection probability; (E), low occurrence and high detection probability; (F), moderate occurrence and high detection probability; (G), high occurrence and high detection probability.

Simulation approach

The occupancy and detection probabilities estimated from the empirical data were used to explore 54 different scenarios for each individual species, using a combination of the number of survey days (occasions: S = (20, 40, 60, 80, 100, 120)) and number of cameras (sites: N = (10, 20, 30, 40, 50, 60, 70, 80, 120)). For each species i, a full detection history was created that is N × S, where each site j from j = 1, 2, …, N is considered to be occupied or not following a Bernoulli process with probability ψi.; we then determined whether a species was detected or not at occupied sites for each occasion t, from t = 1, 2, …, S, following a Bernoulli process with probability pi. In total, 1,000 sets of full detection histories were simulated for each species and each combination of S and N and error was calculated using root mean squared error (RMSE) as: (1) RMSE=Eψˆ−ψ2=Varψˆ+Biasψˆ,ψ2

Given our simulation setting, the generating and estimating model are equivalent, such that bias will generally be low, except for cases of small-sample bias when survey effort is very low. Thus, the majority of error in RMSE across most scenarios is due to the variance, to the extent that the RMSE can often be considered equivalent to the SE(ψ) (Guillera-Arroita, Ridout & Morgan, 2010). For each scenario, the RMSE was plotted for the number of sites (cameras) and the length of the sampling period (days). This was repeated for each of the 13 species. To assess the optimal survey design approach, three different RMSE target values were selected representing differing levels of error; these included RMSE of 0.15, 0.10 and 0.075. It is important to note that an RMSE of 0.15 will result in estimates that will likely be imprecise. This level of error is included for the purposes of comparison, rather than a suggested target value that we recommend should be used in study design. For the purpose of our analysis, we weight occasions (days) and sites (cameras) equally to obtain an optimal solution (but see Table S1). RMSE values of less than 0.1, reflect thresholds used in the wildlife occupancy literature and are considered realistic for determining shifts in occupancy over time and space (MacKenzie & Royle, 2005; Guillera-Arroita, Ridout & Morgan, 2010). The optimal survey design (combination of cameras and number of survey days) was then selected as the minimum survey effort required for each species that enabled the estimate of occupancy to be calculated within the desired level of error. Ideally, the lowest level of RMSE is preferred, but this may be logistically unachievable for some species.

Results

Increasing survey effort reduced RMSE for all species (Fig. 4). However, the optimal combination of the number of sites (motion-activated cameras) and occasions (survey days) varied widely across the 13 species. It was also commonly found that the reduction in RMSE as a result of either increasing the number of cameras or sampling duration would eventually stabilise and would offer limited benefit to further increases in survey effort. Overall, the minimum amount of sampling effort (combination of motion-activated cameras and survey days) required to obtain an acceptable level of error was reduced with increasing daily probability of detection.

Figure 4 Simulation results for each study species.

The influence of survey effort on the error associated with the occupancy estimate (RMSE, root mean squared error), as a function of number of sites (10–120 cameras), occasions (20–120 survey days), and species. Species are presented in order of increasing detection probability (from the top left), with the scale of the y-axis varying between taxa. Raccoons are absent as a reliable estimate could not be achieved due to the lack of data.

Of the species of interest, raccoons had the lowest levels of occupancy coupled with extremely low levels of detection—representing a very rare and hard to detect species in our study area (Fig. 3). Even the maximum survey effort (120 motion-activated cameras operating over 120 days) totaling 14,400 survey days could not guarantee a reliable occupancy estimate for raccoons; approximately 40% of the simulations failed to numerically converge at a maximum-likelihood estimate due to a lack of data. Spotted skunks were also rare and difficult to detect (Fig. 3), however intensive sampling was able to reliably estimate occupancy, requiring 2,000 survey days (e.g., a sampling period of 100 days with 20 cameras) for the highest level of acceptable error (RMSE = 0.15) and 5,000 survey days for the lowest threshold of error (RMSE = 0.075). These results demonstrate the substantial effort that is required to accurately document the presence of rare and elusive species (Fig. 4 and Table 1).

Table 1 Optimal survey design for estimating occupancy with three levels of acceptable error, defined by the root mean squared error (RMSE).

For the purposes of this example we considered the weighting of cameras versus survey days to have the same cost. Raccoons are not included, as a reliable estimate could not be achieved due to the lack of data.

			RMSE 0.15	RMSE 0.10	RMSE 0.075	
Species	Ψ	p	Sitesa× occasions	Total
survey
effort
(days)	Sitesa× occasions	Total
survey
effort
(days)	Sitesa× occasions	Total survey effort
(days)	
Spotted skunk	0.245	0.023	20 × 100	2000	30 × 100	3000	50 × 100	5000	
Elk	0.585	0.024	20 × 80	1600	30 × 120	3600	60 × 100	6000	
Mountain lion	0.600	0.030	20 × 80	1600	30 × 100	3000	60 × 80	4800	
Coyote	0.861	0.031	10 × 60	600	20 × 80	1600	30 × 80	2400	
Bobcat	0.970	0.040	10 × 40	400	10 × 60	600	10 × 80	800	
Gray fox	0.400	0.063	20 × 40	800	30 × 40	1200	50 × 40	2000	
Black bear	0.504	0.072	20 × 40	800	30 × 40	1200	50 × 40	2000	
Virtual sp. 1	0.200	0.120	20 × 20	400	20 × 40	800	30 × 60	1800	
Virtual sp. 2	0.400	0.l60	20 × 20	400	30 × 20	600	50 × 20	1000	
Virtual sp. 3	0.600	0.120	20 × 20	400	30 × 40	1200	50 × 40	2000	
Mule deer	0.925	0.141	10 × 20	200	10 × 20	200	20 × 20	400	
Cottontail rabbit	0.925	0.190	10 × 20	200	10 × 20	200	20 × 20	400	
Notes.

a Sites are the number of cameras and occasions are the number of survey days at each site.

Species that were fairly common with intermediate levels of occupancy but with low detection probabilities (i.e., elk and mountain lion; Fig. 3) also required intensive sampling that used a comparatively large number of both sites and occasions. When the number of occasions increased from 20 to approximately 80 survey days there was a substantial decrease in RMSE for mountain lion and elk, while estimation error was only further improved by including additional sites to the study design (Table 1 and Fig. 4).

For those species with high occupancy (>0.8) and relatively low levels of detection (i.e., coyote and bobcat; Fig. 3), the overall survey effort required to achieve a desired level of error is significantly reduced (compared to spotted skunk, mountain lion and elk). Indeed, increasing the number of occasions at comparatively few sites returns a reliable estimate (Table 1 and Fig. 4). For example, 10 motion-activated cameras proved sufficient to achieve the desired RMSE of 0.15, 0.10 and 0.075 for bobcats, with the reduction in error achieved by including a greater number of survey days (Table 1). As detection probability increases for more common species (i.e., black bear and gray fox), sampling periods over 40 survey days provide no substantial reduction in associated error and argue against continuing the survey. We found the optimal approach for these common species is to sample between 30 and 50 sites (cameras) over a period of 40 survey days depending upon the level of error that is acceptable (see Table 1).

For species with comparatively high levels of daily detection (≥0.12; mule deer, cottontail rabbit and virtual species 1–3), there is only a limited reduction in error associated with lengthening the survey beyond approximately 30 days, particularly for species with moderate to high estimates of occupancy (i.e., virtual species 2, mule deer and cottontail rabbit; Fig. 3). Precise occupancy estimates for these species can be achieved with relatively few cameras (see Table 1). Nonetheless, improving upon these estimates generally requires adding additional sites (cameras) rather than more survey days (Table 1 and Fig. 4). For example, a RMSE of 0.085 can be achieved for cottontail rabbits using only 10 cameras and 20 days of sampling. Further reductions in error cannot be achieved by lengthening the sampling period, with 120 survey days returning an RSME of 0.083 (Fig. 4); however deploying additional cameras can substantially reduce error (Fig. 4). Virtual species 1 and 3 both have low occupancy estimates but comparatively high probabilities of detection (Fig. 3). An intermediate level of survey effort is required in order to achieve the most efficient sampling approach, which depending upon the desired level of precision, balances the number of cameras (20–50) with survey days (20–60; Table 1).

Reducing RMSE by increasing sampling length depends on the probability of detecting the species at an occupied site at least once over the entire sampling duration. We calculated this probability as p* (p* = 1−(1−p)s), where p is the daily detection probability and S is the number sampling occasions. When p* is greater than 0.9 there was very little reduction in RMSE by further increasing the number of sampling occasions (see Table 2; Gerber, Ivan & Burnham, 2014).

Table 2 The probability of capturing at least one image of the study species using different survey designs.

The probability (p∗) of detecting a given species at an occupied site at least once over sampling periods of different durations (10–120 occasions). If p∗ is close to 1 then detection is effectively perfect for that amount of effort. The shading highlights the number of occasions where p∗ ≥ 0.9, and there is only limited improvements in precision to be gained by sampling over longer periods.

Species	Number of sampling occasions	
	10	20	40	60	80	100	120	
Raccoon	0.03	0.06	0.11	0.16	0.21	0.26	0.30	
Spotted skunk	0.21	0.37	0.61	0.75	0.84	0.90	0.94	
Elk	0.22	0.38	0.62	0.77	0.86	0.91	0.95	
Mountain lion	0.26	0.46	0.70	0.84	0.91	0.95	0.97	
Coyote	0.27	0.47	0.72	0.85	0.92	0.96	0.98	
Bobcat	0.34	0.56	0.80	0.91	0.96	0.98	0.99	
Gray fox	0.48	0.73	0.93	0.98	0.99	1.00	1.00	
Black bear	0.53	0.78	0.95	0.99	1.00	1.00	1.00	
Virtual sp. 1	0.72	0.92	0.99	1.00	1.00	1.00	1.00	
Virtual sp. 2	0.72	0.92	0.99	1.00	1.00	1.00	1.00	
Virtual sp. 3	0.83	0.97	1.00	1.00	1.00	1.00	1.00	
Mule deer	0.78	0.95	1.00	1.00	1.00	1.00	1.00	
Cottontail rabbit	0.88	0.99	1.00	1.00	1.00	1.00	1.00	

Our simulation results reveal broad patterns in survey design when using motion-activated cameras that depend upon how easy a species is to detect and how common it is across the landscape (Fig. 5). Rare species with low detection require an intensive sampling approach that combines multiple camera sites and occasions to reliably calculate an occupancy estimate, whereas the best strategy for more common species with low levels of detection involves increasing the number of survey days (occasions) at comparatively few sites (≤30 cameras). As detection probability increases, the overall survey effort required to achieve an acceptable level of precision in occupancy is reduced. Species that exhibit moderate detectability but remain comparatively rare generally require an intermediate number of cameras and greater survey lengths to improve precision, compared with common and detectable species that can be surveyed precisely with relatively few sites and short sampling periods (Fig. 5).

Figure 5 Broad recommendations on survey design for studies exploring occupancy using motion-activated cameras.

The symbols indicate high (+), intermediate (O) and low (−) amounts of effort, for the relative number of cameras and survey days to achieve an optimal survey design. From the upper-right to the lower-left, an increasing amount of survey effort is required to reliably estimate occupancy.

Discussion

Reliable indicators that document landscape-to-regional biodiversity are urgently needed given the global extinction crisis (Mace & Baillie, 2007; Butchart et al., 2010). Motion-activated cameras can provide scientists and wildlife managers with a very powerful tool for recording the presence and occupancy of a diverse range of species (O’Connell & Bailey, 2011; Ahumada, Hurtado & Lizcano, 2013), particularly given the advances in storage, reliability and battery life of the latest devices (Jamie, 2012). Nevertheless, in common with other ecological monitoring and research programs, successful sampling strategies rely on detailed study design. Indeed, pursuing an optimal survey design allows available time and resources to be most efficiently used, while also providing guidance as to whether the objectives are achievable and justified given potential funding constraints (McDonald-Madden et al., 2010).

Survey design using an occupancy framework has been explored in a number of primary research papers, which given particular detection and occupancy values have provided detailed theoretical insights into the trade-off between the number of sites and the number of occasions (MacKenzie & Royle, 2005; Bailey et al., 2007; Guillera-Arroita, Ridout & Morgan, 2010; Guillera-Arroita & Lahoz-Monfort, 2012). Furthermore, a number of excellent simulation tools are freely available for exploring survey design (e.g., GENPRES: Bailey et al., 2007; SODA: Guillera-Arroita, Ridout & Morgan, 2010). It is important to note that the purpose of our study was not to advance these analytical methods, or indeed provide a detailed statistical explanation of the simulation results. Instead, we have used an occupancy framework to specifically analyse a motion-activated camera dataset to provide a range of realistic scenarios that outline how survey design and effort varies depending upon the species of interest. Although a comparatively straightforward approach from a quantitative perspective, we believe that the value of our study lies in providing guidance and key worked examples to a broad audience of wildlife practitioners that may be planning to use cameras for monitoring or research initiatives. Furthermore, cameras present a useful tool for surveying rare species across the landscape, and as such, the inclusion of species with very low detection and occupancy demonstrate the level of effort required to achieve desired objectives for these taxa. Our study follows on from recent research that has explored camera survey design in terms of species richness (Si, Kays & Ding, 2014) and time-triggered devices (Hamel et al., 2013).

We found substantial differences in optimal survey designs across mammal species from our study area, concurring with MacKenzie & Royle (2005) who highlighted that surveying as many sites as possible is not the most efficient approach to reducing overall occupancy estimation error. Instead, obtaining a reliable and efficient occupancy estimate requires tailoring the study design to the species of interest. Unsurprisingly, the most challenging taxa to develop an appropriate survey design for were the rare and hard to detect species (e.g., raccoons). Even with considerable survey effort it was challenging, if not impossible, to reliably estimate occupancy based on our criteria of RMSE. If the goal of a study is to estimate the occupancy of a rare species that is difficult to detect, it may be necessary to reposition the cameras to target specific taxa (Karanth & Nichols, 2002), use baits or lures (Thorn et al., 2009) or employ multiple methods (e.g., cameras, sign surveys: Magoun et al., 2011). Even if each method individually has a low probability of detection, the combined effect of all methods incorporated together will be greater, and thus potentially lead to a reliable occupancy estimate. Such an approach can be carried out using multi-scale occupancy models, which allow data to be incorporated from multiple detection methods while permitting estimation of occupancy across different spatial scales (Nichols et al., 2008).

Alternatively, for threatened and endangered species it may be more appropriate to forego estimating species occurrence and simply try to determine if the species is present in the area of interest (MacKenzie et al., 2006; Si, Kays & Ding, 2014). We provide an example in the Supplemental Information evaluating the probability and financial cost of photographing a very rare species (ψ = 0.05 and p = 0.05) at least once (see Table S2 and Spreadsheet S1). Meanwhile, for common and highly detectable species (e.g., cottontail rabbits and mule deer), relatively few motion-activated cameras and survey days are necessary to provide accurate and precise occupancy estimates. As such, rapid assessment surveys could be routinely used to monitor these species relatively inexpensively, while taxa that are comparatively rare across the landscape but yet remain highly detectable (e.g., virtual species 1) require greater survey effort, and are therefore best sampled using an intermediate number of sites and survey days.

Our results demonstrated that conducting extended survey periods to estimate occupancy for species with moderate to high detection probability may not reduce error despite continued effort, while also potentially leading to issues associated with the violation of closure (MacKenzie et al., 2006). Indeed, deriving a biologically meaningful sampling period (e.g., season) during which occupancy status is assumed not to change may vary depending upon the target species and research question, and is therefore a fundamental consideration for survey design (see Gerber, Williams & Bailey, in press). Furthermore, trade-offs in survey approach will be necessary for community-level research, as it is unlikely that a single design will be most efficient for all species. One potential way forward is to initially define the season (timing and scope of inference), and then consider all the species of interest that can be reasonably detected and use the optimum survey effort required to detect all of these taxa (Si, Kays & Ding, 2014).

The species included in our study cover a diverse range of daily detection probabilities and occupancy estimates representing a broad spectrum of mammals. The findings can therefore be applied to other taxa and ecosystems where cameras are being used to study small to large terrestrial mammals. To determine an optimal study design, we suggest that researchers first investigate the occupancy (i.e., common, moderately common, rare) and detection (i.e., low, moderate, high) characteristics of their target species. Table 1 and Fig. 5 can then be used to guide the required sampling effort (number of sites and survey days) for an acceptable level of error (see also tables provided in MacKenzie & Royle, 2005; Guillera-Arroita, Ridout & Morgan, 2010). For many mammals, there may already be published literature in which species occupancy and detection probabilities could be obtained. If no prior information is available, short pilot studies can be very effective in obtaining values for occupancy and detection probability, particularly as these values are often highly site specific, or species experts could be consulted to give a rough estimate regarding occupancy and detection probabilities, depending on body size, behaviour and ranging patterns. Additionally, a recent study has demonstrated a two-stage Bayesian method for incorporating uncertainty in the initial estimates employed in study design (Guillera-Arroita, Ridout & Morgan, 2014).

Finally, as survey effort involves a trade-off between cameras and sampling length, it is important to note that the financial costs associated with these different scenarios may vary considerably. For example, surveying additional sites may require purchasing more cameras, while increasing the survey duration may require personnel to make additional site visits to keep the cameras functioning properly. Under certain scenarios where the number of sites is limiting the accuracy/precision of the estimate and there is sufficient time for two surveys to be conducted within a designated season, cameras can be set for the necessary period and then moved (e.g., Karanth & Nichols, 2002). Interestingly, this has also proved an efficient method for measuring species richness (Si, Kays & Ding, 2014). It is important to bear in mind that the optimal solution will depend upon the costs associated with camera operation and maintenance versus the costs of procuring cameras (see Table S1 and Spreadsheet S1 for further details on survey design and financial cost).

In conclusion, our study investigates the optimal survey effort (sites vs. occasions) required for determining occupancy with a desired level of error across a range of mammalian species using motion-activated cameras. The results of our simulation approach clearly highlight that simply increasing survey effort is not the most efficient strategy for obtaining a reliable occupancy estimate. The guidelines presented in the paper are based on the analysis of an empirical dataset to provide a real world example that does not solely rely on the simulation of virtual data, while still being directly applicable to research and monitoring programs conducted in other terrestrial ecosystems. We emphasize the use of biological knowledge of the target species coupled with clearly defined a-priori objectives that link monitoring or research effort with defined ecological questions or conservation actions (Martin et al., 2009). Our study also illustrates the value of data simulation approaches for assessing methods and study design before embarking on empirical data collection (Zurell et al., 2010; Ellis, Ivan & Schwartz, 2013). However, it is not always feasible for practitioners to carry out their own simulation exercises. Thus, research such as ours that can provide broad guidelines when a species of interest can be generally classified as rare or common and easily or difficult to detect will be of great utility to designing effective studies.

Supplemental Information

Table S1 Survey design as a function of cost and effort

Optimal survey design for estimating occupancy with three levels of acceptable error defined by the root mean squared error (RMSE). In the main analysis (Table 1) we considered the costs of cameras and occasions to be equivalent and determined optimality solely on the minimum number of survey days (sites × occasions). Here we provide a scenario that also considers the financial costs of conducting a motion-activated camera survey, with lower cost survey designs that meet the desired level of precision being selected in preference over more expensive approaches. For the purpose of our scenario, a motion-activated camera = 250, while the cost of surveying an occasion at = 10, which incorporates the cost of personnel, vehicle use, batteries and maintenance of the equipment. See Spreadsheet S1 to calculate alternate scenario costs.

Click here for additional data file.

Table S2 Probability and associated cost of capturing a single image of a very rare species

The probability (p∗∗) of capturing a photograph of a very rare species (ψ = 0.05 and p = 0.05) as a function of the number of occasions and sites (a), where (p∗∗ = 1−[1−ψ(1−(1−p)S)]N). The survey cost in dollars calculated on the basis that each camera is 250 and a survey occasion costs 10 (b). See Spreadsheet S1 to calculate alternate scenario costs.

Click here for additional data file.

Table S3 Survey cost tool

The spreadsheet allows the financial costs of different scenarios to be calculated depending upon the cost of each camera and occasion (e.g., field costs incurred per day for each camera).

Click here for additional data file.

Table S4 Mule deer in Colorado, USA

For consideration as an image to be used on the peerJ website. Credit: Graeme Shannon. Manuscript 2014:06:2254:1:0:NEW The mule deer is one of the species that we explore in our camera trap study and represents our research.

Click here for additional data file.

We thank L Sweanor, B Dunne, and K Logan for their invaluable assistance in the field and the landowners who allowed us access to their properties for our research. We are grateful to K Crooks and L Bailey for comments on an earlier version of the manuscript and we would like to thank the editor, P Gandini, and two anonymous reviewers for providing constructive suggestions that greatly improved the manuscript.

Additional Information and Declarations

Competing Interests

Author Contributions

Animal Ethics

The authors declare that there are no competing interests.

Graeme Shannon conceived and designed the experiments, analyzed the data, wrote the paper, prepared figures and/or tables, reviewed drafts of the paper.

Jesse S. Lewis conceived and designed the experiments, analyzed the data, prepared figures and/or tables, reviewed drafts of the paper, designed and executed the motion-activated camera study.

Brian D. Gerber analyzed the data, contributed reagents/materials/analysis tools, prepared figures and/or tables, reviewed drafts of the paper.

The following information was supplied relating to ethical approvals (i.e., approving body and any reference numbers):

As the study involved non-invasive sampling using motion-activated cameras there was no requirement for institutional review of the proposed research.

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
