# Peer review of "Recommended survey designs for occupancy modelling using motion-activated cameras: insights from empirical wildlife data"

_PeerJ, doi:10.7717/peerj.532_

## Round 0.1 · original submission · Major Revisions

The paper is well written and the application of the methods is correct however authors will have to pay special attention clarifying how their work brings an advance to previous work in the area.

Take special attention to comments made by Reviewer 1:
1) why the approach provide an advance by using empirical data when such data are used in selecting scenarios to evaluate in the simulations
2) Include and discuss ref cited by Reviewer 1. 3) clearly state, describe and explain the rational for replication
4) Other camera studies could have another cost when used other camera models and survey conditions change, include a sentence to clarify it.
5) Some detailed edits are included

Reviewer 1 ·

Basic reporting

The writing is generally clear across the manuscript. There are however places where wording should be more precise (I list these below). In terms of setting the work in the context of relevant literature I found two gaps that seem particularly relevant here and which I think should be covered:

1) Continuous vs discrete sampling protocol: Camera traps produce data in continuous time and, although frequently done, cutting time into intervals is somewhat arbitrary. Guillera-Arroita et al (2011) covers optimal design for occupancy-detection models based on continuous sampling protocols (such as camera trap data). The authors should clarify that they have chosen to consider discretization of the data but that alternative modeling methods are available. Again, the continuous models can be mentioned in the discussion around P11, L26-32.

2) Uncertainty in initial parameter values: The authors use estimates obtained from current data to inform the design of future surveys, however they disregard uncertainty in those estimates, and do not discuss anywhere the circularity involved in the fact that study design requires initial values of parameters that are the very same object of estimation. Guillera-Arroita et al (2014) present ways to deal with this circularity (ways of incorporating uncertainty in the initial estimates). Also, one of the methods proposed is to follow a 2-stage approach so this work could also be relevant for the discussion, where the authors talk about pilot studies (P12,L9)

Refs:
- Guillera-Arroita G, Morgan BJT, Ridout MS, Linkie M (2011) Species occupancy modeling for detection data collected along a transect. Journal of Agricultural, Biological and Environmental Statistics 16: 301-317.
- Guillera-Arroita G, Ridout MS, Morgan BJT (2014) Two-stage Bayesian study design for species occupancy estimation. Journal of Agricultural, Biological and Environmental Statistics 19: 278-291.

Detailed edits:
• P2,L21-21: “maximizing” is not the right word here.
• P7,L11: note that in the abstract 10-120 refers to sites and 20-120 to occasions. Here it is the other way around.
• P7,L30-31: “considered realistic for determining…”, by whom? I could not find in the cited papers any reference to setting RMSE as high as 0.15 as targets for design…
• P8,L7: remove “generally”, increasing survey effort will always reduce RMSE (even if only marginally).
• P8,L9-11: This observation does not seem correct. Increasing the number of sites will never “stabilize” RMSE. Note that the variance decreases with a factor 1/nsites, so it always decreases at the same rate with the number of sites.
• P8, L25: “maximized’ does not seem the correct word here.
• P8, L22: remove “in general”
• P10,L14: again “maximized” does not seem the correct word here. Alternatives could be “to be best used”, “to be most efficiently used”, etc…
• P12,L6-7: The authors should mention the tables and software already available in the literature for this purpose (tables: MacKenzie & Royle 2005; Guillera-Arroita et al 2010; software: see next comment)
• P13,L6: you should mention here that simulation tools specifically used for this purpose are freely available (Genpres by Bailey et al 2007, SODA by Guillera-Arroita et al 2010, and R-scripts to conduct the power analysis in Guillera-Arroita & Lahoz-Monfort 2012).
• Table 1: add CIs for the estimates
• Table 1: note that the raccoon is not included.
• Table 2: add to the caption a clarification that when p* is close to 1 it means that detection is practically perfect with that amount of effort

Experimental design

The methods seem to be correctly applied. One comment is that a RMSE of 0.15 seems quite large (it leads to CIs of length about 0.6, that is, covering most of the range of probabilities; corresponding estimates would be very imprecise). I would stress this point in the manuscript before people start taking this as a target for study design (or perhaps I would stick to smaller RMSE values).

Validity of the findings

My main concern with this article is that, while the application of methods is correct, it is not clear how it brings an advance with respect to previous work in this area. I understand that PeerJ does not worry with “degree of advance”, “novelty”, “impact”, etc… however it also notes that replication of accepted results is not allowed. Unfortunately, I think that this paper is relatively close to “replication” of previous work. I think the editor will be the best person to judge whether the contribution is suitable for the journal, and therefore I leave this decision to him/her. I provide an objective description of the situation below.

In this paper the authors use simulations to assess the RMSE obtained from a constant occupancy-detection model as a function of sampling sites and number of replicates visit to identify the optimal design (that which achieves a target RMSE with minimum total effort). The same thing has already been done before (e.g. Bailey et al 2007, Guillera-Arroita et al 2010), plus has also been done using asymptotic approximations instead of simulations (MacKenzie & Royle 2005, Guillera-Arroita et al 2010). The authors suggest that their approach provide an advance because they use “empirical data”. However, the only use of such data is to select the scenarios to evaluate in the simulations (combination of occupancy and detectability parameters to simulate). The papers stated above explore a whole range of parameter combinations and some provide software to explore any scenario a user wants to assess. The results/conclusions obtained in the present article do not seem to be new. The study is not specific about camera traps (the empirical data used to determine the parameter values to use in the simulations come from camera traps, but the study design analysis is general).

Reviewer 2 ·

Basic reporting

P4. L18-19
The authors state correctly state that replication is a necessary property of occupancy studies leading to a trade-off between the number of sites and within-site replication. However, the way this is phased makes it sound as if this is a specific property of occupancy studies rather than a general property of population sampling.

Experimental design

No Comments

Validity of the findings

P10. L20-29
This paragraph is devoted to alternative designs for rare and difficult to detect species (e.g. the use of multiple techniques) to increase detection probability. The use of lures or baits is not mentioned in this paragraph even though this has the potential to increase detection probability. The use of lures is mentioned in the subsequent paragraph but only in relation to the determination of the presence or absence of a rare species in an area. It is therefore unclear if the authors believe that the use of lures is appropriate for use in occupancy studies or not, and if not why not?

P11
This page is principally devoted to a discussion of the relative costs of different survey designs (with more detail given in tables S1 and S2). A consideration of survey cost is important, however in this case it is based on a restricted set of assumptions about camera cost ($250) and the cost of a survey occasion ($10). My concern here is that whilst these may be valid for the particular circumstances encountered by the authors, they are they are likely to be of limited use to other researchers as they will be using different camera models (with different purchase costs and different requirements for the duration between revisits) and will have different costs for site visits, particularly given that cameras are often used as a survey tool for animals in remote locations (there seems to be an assumption that site visits are relatively cheap (P11, L17)). It would be interesting to explore a number of scenarios related to number of sites vs the number of occasions, but as this is not the main scope of the paper, the limitations should be more clearly articulated or this section omitted.

Additional comments

No Comments

---

## Round 0.2 · accepted · Accept

You have addressed the major points suggested by both reviewers and the paper is clearer now. Your study will provide a clear guidance to wildlife practitioners using this survey techniques and for sure will be helpful to Field work of many Conservation Biologists.